# Birth weight, growth, nutritional status and mortality of infants from Lambaréné and Fougamou in Gabon in their first year of life

**Rella Zoleko-Manego**[1,2,3], **Johannes Mischlinger**[3], **Jean Claude Dejon-Agobé**[1,2,4], **Arti Basra**[1], **J. Rodolphe Mackanga**[1], **Daisy Akerey Diop**[1], **Ayola Akim Adegnika**[1,2,5,6], **Selidji T. Agnandji**[1,2], **Bertrand Lell**[1,7], **Peter G. Kremsner**[1,2,5,6], **Pierre Blaise Matsiegui**[8], **Raquel González**[9,10,11], **Clara Menendez**[9,10], **Michael Ramharter**[1,3], **Ghyslain Mombo-Ngoma**[1,2,3]*

**1** Centre de Recherches Médicales de Lambaréné (CERMEL), Lambaréné, Gabon, **2** Institut für Tropenmedizin, Universität Tübingen, Tübingen, Germany, **3** Department of Tropical Medicine, Bernhard Nocht Institute for Tropical Medicine & I. Dep. of Medicine, University Medical Center Hamburg-Eppendorf, Hamburg, Germany, **4** Division of Internal Medicine, Center of Tropical Medicine and Travel Medicine, Department of Infectious Diseases, Amsterdam University Medical Centers, University of Amsterdam, Amsterdam, The Netherlands, **5** German Center for Infection Research (DZIF), African Partner Institution, CERMEL, Gabon, **6** German Center for Infection Research (DZIF), Partner Site Tübingen, Germany, **7** Division of Infectious Diseases and Tropical Medicine, Department of Medicine 1, Medical University of Vienna, Vienna, Austria, **8** Centre de Recherches Médicales de la Ngounié (CRMN), Fougamou, Gabon, **9** Barcelona Institute for Global Health, ISGlobal, Hospital Clínic - Universitat de Barcelona, Barcelona, Spain, **10** Consorcio de Investigación Biomédica en Red de Epidemiología y Salud Pública (CIBERESP), Madrid, Spain, **11** Faculty of Medicine. Central University of Catalonia (UVic-UCC), Can Baumann, Barcelona, Spain

* ghyslain.mombongoma@cermel.org

**Data Availability Statement:** The data are available at DOI: 10.5281/zenodo.4244987.

## Abstract

### Background

Malnutrition and low birth weight (LBW) are two common causes of morbidity and mortality among children in sub-Saharan Africa. Both malnutrition and LBW affect early childhood development with long term consequences that may vary in their degree depending on the geographical setting. This study evaluates growth, nutritional status and mortality of infants from Lambaréné and Fougamou in Gabon from a birth cohort of a malaria in pregnancy clinical trial (NCT00811421).

### Method

A prospective longitudinal birth cohort conducted between 2009 and 2012, included infants that were followed up from birth until their first-year anniversary. The exposure of interest was low birth weight and the outcomes explored were growth represented by weight gain, the nutritional status including stunting, wasting and underweight, and the mortality. Scheduled follow-up visits were at one, nine and 12 months of age. Logistic regression was used to assess the association between low birth weight and growth and nutritional outcomes, and cox regression was used for mortality.

**Funding:** The main (MIPPAD) study was funded by the European Developing Countries Clinical Trials Partnership (EDCTP; IP.2007.31080.002). The funder had no role in study design, data collection and analysis, decision to publish, or preparation of the manuscript.

## Result

A total of 907 live-born infants were included in the analysis. The prevalence of LBW was 13% (115). At one month of life, out of 743 infants 10% and 4% presented with stunting and underweight, respectively, while these proportions increased at 12 months of life to 17% and 21%, respectively, out of 530 infants. The proportion of infants with wasting remained constant at 7% throughout the follow-up period. Stunting and underweight were associated with LBW, adjusted odds ratio (aOR): 2.6, 95% confidence interval (95%CI): 1.4–4.9 and aOR: 4.5, 95%CI: 2.5–8.1, respectively. Preterm birth was associated with stunting, aOR: 2.7, 95%CI: 1.2–6.3 and underweight, aOR: 5.4, 95%CI: 1.7–16.1 at one month of life. Infants with LBW were at higher hazard of death during the first year of life, adjusted hazard ratio 4.6, 95%CI: 1.2–17.0.

## Conclusion

Low birthweight infants in Gabon are at higher risks of growth and nutritional deficits and mortality during the first year of life. Tailored interventions aiming at preventing adverse pregnancy outcomes including LBW, early detection and appropriate management of growth, and nutritional deficits in infants are necessary in Gabon.

## Introduction

Malnutrition is a public health concern particularly in economically disadvantaged regions of the word. According to the 2017 Food and Agriculture Organization's report, about 821 million people suffer from various forms of malnutrition in the world [1]. Asia and Africa harbor the highest numbers of individuals suffering from malnutrition with children being the most affected [2]. In this particular population, malnutrition is reported as one of the most common causes of morbidity and mortality, mostly affecting early childhood development. Long term consequences of malnutrition include increased risk of diet-related non-communicable diseases such as diabetes and hypertension [3–5]. However, the impact of malnutrition may vary depending—amongst many factors—on the socioeconomic and geographical settings, the burden of infectious diseases, and the availability and capacity of the health care system in respective regions [6–9].

Stunting, wasting and underweight are three established indicators for the nutritional status of infants and children, indicating their overall health and well-being [10, 11]. Stunting, expressed in height-for-age, is reported to be a strong marker of unhealthy growth [12]. It is the most prevalent form of child malnutrition. It is a slow, cumulative process developing over a long period, it is the primary manifestation of malnutrition in early childhood. Once established, stunting and its effects typically become permanent [13]. Stunting is an appropriate indicator for chronic malnutrition. Wasting, using weight-for-height and underweight, using weight-for-age are valuable indicators for acute malnutrition with potential for reversal. Underweight is most commonly used as a nutritional indicator due to difficulties in measuring height in health programs implemented in low- and middle-income countries [14]. In this context and despite its limitation, weight alone appears to be the indicator most often used by parents and primary health care personnel to quickly estimate the growth in infancy [15].

While stunting, wasting and underweight clearly refer to malnutrition during childhood, low birth weight (LBW) may be regarded as an indicator for malnutrition during pregnancy.

LBW is defined as birth weight less than 2500 g and is recognized as an indicator of preterm birth or intrauterine growth retardation. Children born with LBW remain at risk of malnutrition [16, 17] and are therefore of special interest for growth monitoring. Furthermore, the association between LBW and malnutrition and the subsequent increased mortality constitute the reason for monitoring child growth during the first 1000 days of life [18, 19]. Compared to those born with normal weight, the World Health Organization (WHO) estimates an increase of 2-to-8 fold the death rate among children born with LBW during their first year of life [20]. The impact of LBW on nutritional indices and survival may however vary according to the setting in which the child lives [18, 21, 22].

There is a lack of data on infant and child growth for many countries in Africa, particularly for the Central African region where the prevalence of LBW is high [23]. Most of the previous studies have focused on the determinants of LBW with the aim to identify modifiable factors while the outcome of infants after birth was rarely investigated. Moreover, there is a lack of prospective studies in Africa focusing on the nutritional status of infants in general and infants born with LBW in particular. Taking advantage of a clinical trial conducted in Gabon on pregnant women and their offspring, this analysis evaluates the nutritional status, growth and mortality of infants in a prospective birth cohort during their first year of life based on their birth weight. This study therefore provides important information about the impact of LBW on nutritional deficits in the first year of life and survival of infants in a semi-urban region of central Gabon.

## Methods

### Study site and population

This study was conducted in Lambaréné and Fougamou, two semi-urban areas located in the central region of Gabon, central Africa. Study participants were infants born from HIV-negative pregnant women included in a clinical trial assessing intermittent preventive treatment of malaria during pregnancy (Malaria in Pregnancy Preventive Alternative Drugs [MiPPAD]; clinical trials identifier: NCT00811421 [24]). Live-born infants from singleton pregnancies with weight reported at birth were included in this analysis.

### Study design

This is a birth cohort study of infants born in the MiPPAD trial. Details of recruitment and data collection procedures have been described elsewhere [24]. Briefly, HIV-uninfected pregnant women were recruited before their third trimester. Eligible pregnant women were randomly allocated to receive two doses one month apart of intermittent preventive treatment during pregnancy (IPTp) with either sulfadoxine/pyrimethamine or mefloquine. Enrolled pregnant women were then followed up until one month after delivery. At birth, demographic data and anthropometric parameters of the newborn were recorded and a physical examination was performed. Infants were followed up until their first anniversary with scheduled visits at one, nine and 12 months of age at which anthropometric measures were recorded including weight and height. When the visits at the health facility were missed, home visits were organized whenever possible.

### Study procedures and variables

The study was conducted from September 2009 to April 2012. Mother's baseline characteristics were recorded at recruitment including maternal age and literacy. Maternal age was calculated from the date of birth recorded in a health booklet or by self-reported date of birth. Maternal

age was categorized as following: adolescent, aged < 20 years; young adults, aged 20–35 years and older adults, aged > 35 years. Literacy of the mother was defined as the ability to read or write. Newborn delivery characteristics recorded included place of birth, gestational age, weight, height, infant sex and congenital abnormalities. For each included newborn, weight was captured within a few hours after birth using a weekly re-calibrated electronic pediatric weight scale. For those with no weight available at birth but in the first week of life in case of home deliveries weight was estimated using a linear regression model [24]. The height of the infant was measured in centimeters using a calibrated gauge with a fixed headrest and a movable footrest perpendicular to the surface of the table placed in contact with the infant's feet for measurement in lying position. Gestational age was calculated using modified Ballard score at birth [25]. Prematurity was defined as a gestational age at birth less than 37 weeks. Z-score for Weight-for-age, weight-for-height and height-for-age were calculated using standard formula [26]. Height-for-age Z-score (HAZ, <-2 standard deviations below an international reference mean), weight-for-age Z-score (WAZ <-2 SD), and weight-for-height Z-score (WHZ <-2 SD) were used to define respectively stunting, underweight and wasting.

## Statistical methods

Data were collected using a paper case report form and digitalized using OpenClinica software. Double-entry of data was performed to ensure the reliability of data. The clean database was extracted for statistical analysis. Stata IC/V.13.1 for Windows (StataCorpLp, College Station, Texas, USA) was used to perform the statistical analysis. Because a high proportion of infants were dropped from the subsequent visits at month 9 and month 12, a sensitivity analysis was performed that consisted on comparing baseline characteristics of infants attending each visit to explore whether any differences existed that could be associated with loss to follow up. The main exposure variable was LBW defined as weight less than 2500 g at birth. Weight was used to characterize an infant's growth during a one-year period. During the analysis, two study groups were considered, LBW and normal birth weight (NBW) groups and the results presented accordingly. Categorical variables were summarized as counts and proportions and were compared using the chi square test while continuous variables were summarized by means and standard deviation (SD) and compared using the Z-test. Weight gain ratio was presented as the ratio of the weight of infants at the actual visit divided by the weight of the infants at the previous visit. Malnutrition determinants were assessed at different study time points considering birth weight categories (LBW vs NBW). Assessment of malnutrition at month 9 and month 12 were done only for participants present at the month 1 visit. The main study was designed as a randomized clinical trial with the assumption that the confounders will be distributed between the treatment arms and therefore not systematically recorded. Variables such as parity, trimester of first antenatal clinic visit, BMI and MUAC, and infectious diseases were collected from the mother and they have been controlled for in the analyses of the determinants of LBW. For the current analysis of the effect of LBW on growth, nutrition and mortality, the maternal variables retained to be controlled for are age and literacy as they are considered to potentially affect the way mothers provide care and follow nutrition instructions for their infants. The infant's sex has been taken as a forced variable as sex and age are common confounding factors. Several variables including mother's age and literacy, infant's sex and prematurity are recognized as risk factors for malnutrition and infant death [10, 27]. Logistic regression was used for multivariate analysis of risk factors associated with malnutrition. Mortality was calculated as the number of deaths per 1000 live births. Time-to-event analysis was conducted using a Kaplan-Meier survival analysis with 'death' defined as an event of interest. Plots were created to visualize survival curves stratified by birth weight. Log rank test

was used to compare the survival functions between participants born with LBW and those born with normal weight. Lost to follow-up participants were included in the analysis and censored at their last contact. Hazard of death was assessed by Cox proportional hazards model. Crude and adjusted hazard ratios and associated 95% confidence interval (95%CI) were generated. For each statistical analysis, the level of statistical significance was set at p-value less than 0.05.

## Ethical consideration

The clinical trial was approved by the Comité d'Ethique Régional Indépendant de Lambaréné (CERIL) and the Ministry of Health in Gabon and was conducted in line with the Good Clinical Practice (GCP) principles of the International Conference on Harmonization and the Declaration of Helsinki.

## Results

### Study population characteristics

As depicted in Fig 1, a total of 983 live delivered newborns from the MIPPAD trial in Gabon were recorded. Thirty multiple gestations and 46 newborns with missing birth weight records were excluded from this analysis, giving a total of 907 infants included. Their characteristics are described in Table 1. Of the 907 live births, the prevalence of LBW was 12.7% (115). There were 50 (5.5%) preterm newborns and they endured the highest prevalence of LBW of 38% (19/50). There were more LBW infants from adolescent mothers 17.4% (50/287) than from mature adult mothers, 7.0% (5/71). The mean gestational age at birth was 40 weeks (SD: 2.0). During the follow-up, 765 (84.3%) attended the 1-month visit, 575 (63.4%) attended the 9-month visit, and 576 (63.5%) attended the 12-month visit including 64 infants that missed the 9-month visit (Fig 1). Taking into account the participants' drop out, there were significantly more dropouts in the LBW group at the 9-month visit and at 12-month visit there were significantly more dropouts from the literate mothers group and from the younger mothers groups (S1 Table).

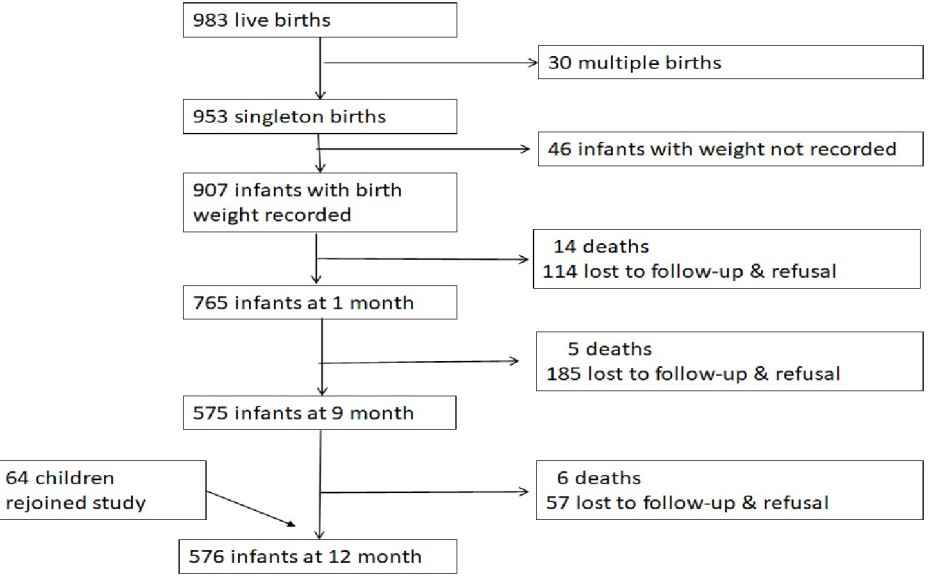

**Fig 1. Flow of children in the study from birth until 12 months of age.**

**Table 1. Distribution of Low birth weight according to maternal and newborn characteristics at delivery.**

| | | N | Low Birth Weight n (%) | Normal Birth Weight n (%) | *p-value* |
|---|---|---|---|---|---|
| **Place of birth** | | | | | |
| | Maternity | 827 | 105 (12.7) | 722 (87.3) | 0.9 |
| | Home | 80 | 10 (12.5) | 70 (87.5) | |
| **Preterm birth*** | | | | | |
| | Yes | 50 | 19 (38.0) | 31 (62) | <0.001 |
| | No | 832 | 89 (10.7) | 743 (89.3) | |
| **Infant Sex*** | | | | | |
| | Male | 454 | 53 (11.7) | 401 (88.3) | 0.3 |
| | Female | 442 | 60 (13.6) | 382 (86.4) | |
| **Mother Literacy** | | | | | |
| | Literacy | 745 | 92 (12.4) | 653 (87.6) | 0.5 |
| | Illiteracy | 162 | 23 (14.2) | 139 (85.8) | |
| **Maternal age (year)** | | | | | |
| | <20 | 287 | 50 (17.4) | 237 (82.6) | 0.009 |
| | 20–35 | 549 | 60 (10.9) | 489 (89.1) | |
| | >35 | 71 | 5 (7.0) | 66 (93) | |
| **Congenital Abnormalities*** | | | | | |
| | Present | 22 | 2 (9.1) | 20 (90.9) | 0.6 |
| | Absent | 879 | 110 (12.5) | 769 (87.5) | |

N: number, (): % percentage,

*missing data.

## Growth or weight gain during the 12 months follow-up

As described in Table 2, the mean weight was 2994 g, 4272 g, 8276 g, and 9003 g at birth, 1, 9 and 12 months of age, respectively. The mean birth weight was significantly lower in the LBW group and the difference in mean weight remained as such during the entire follow-up period (Table 2). However, the weight gain ratios were similar in both groups, 1.4 versus 1.6 at one month, 1.9 versus 2.2 at nine months, and 1.1 versus 1.1 at 12 months for the normal birth weight group and the LBW group, respectively (Fig 2).

## Malnutrition status at follow-up visits and impact of low birth weight

The prevalence of stunting was 9.7% (72/743; 95%CI: 8–12), 9.5% (50/524; 95%CI: 7–12) and 16.8% (89/530; 95%CI: 14–20) at 1, 9 and 12 months of age, respectively. As presented in

**Table 2. Mean of weight at birth, 1, 9 and 12 month visits according to birth weight.**

| | Normal birth weight | | | Low birth weight | | | Total | | | P value |
|---|---|---|---|---|---|---|---|---|---|---|
| | n | Mean | SD | n | Mean | SD | N | Mean | SD | |
| Birth | 792 | 3113 | 366 | 115 | 2172 | 353 | 907 | 2994 | 480 | <0.0001 |
| Month 1 | 674 | 4369 | 619 | 91 | 3550 | 934 | 765 | 4272 | 719 | <0.0001 |
| Month 9 | 515 | 8328 | 1105 | 60 | 7828 | 1008 | 575 | 8276 | 1106 | 0.0004 |
| Month 12 | 509 | 9058 | 1181 | 67 | 8587 | 1321 | 576 | 9003 | 1207 | 0.001 |

SD: standard deviation, n: population.

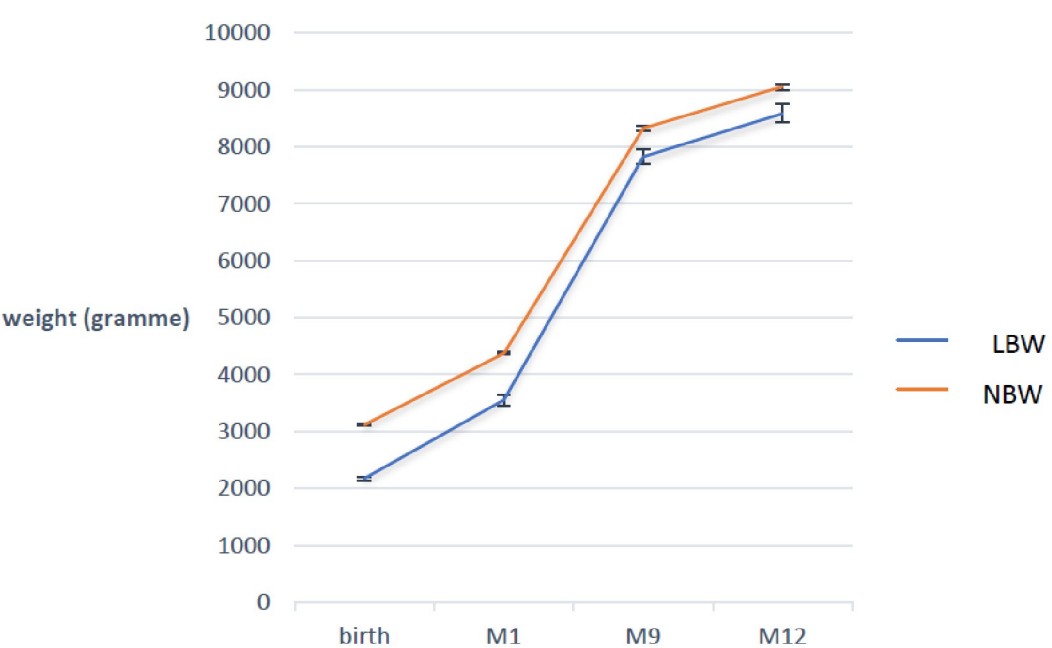

**Fig 2. Mean (SE) changes in weight at birth, 1, 9 and 12 months by birth weight.** Abbreviations: M1: month one; M9: month nine; M12: month twelve, LBW: Low Birth Weight; NBW: Normal Birth Weight.

Table 3, stunting was significantly more prevalent in infants with LBW from month 1 to month 12 compared to their normal birth weight counterpart. Preterm birth was found to be a factor associated with stunting at month 1 while being a female was associated with a significantly lower risks of stunting at 9 and 12 months of age (Table 3). The prevalence of wasting was 5.8% (43/743; 95%CI: 4–8), 8.2% (43/524; 95%CI: 6–11) and 7.0% (37/530; 95%CI: 5–10) at month 1, 9, and 12, respectively, and none of the preterm born infants was reported with wasting. Mother's illiteracy was found to be associated with wasting at months 9 and 12. The prevalence of underweight was 4.0% (30/743; 95%CI: 3–6), 14.5% (76/524; 95%CI: 12–19) and 20.7% (110/530; 95%CI: 7–24) from months 1, 9 to 12, respectively. There was significantly more underweight observed among infants born small compared to their normal birth weight counterparts. Mother's illiteracy was found to be a factor significantly associated with underweight at months 9 and 12, respectively, while male gender of infants was also an associated factor but at month 12 only (Table 3).

After adjusting for preterm birth, infant sex, maternal age and literacy, LBW remained an independent risk factor associated with higher odds of stunting and underweight with strong statistical evidence at month 1 and month 12 (adjusted OR (aOR) 10.3, 95%CI:5.9–17.9; aOR 33.1, 95%CI:12.2–83.2 and aOR 2.6, 95%CI:1.4–4.9; aOR 4.5, 95%CI:2.5–7.2; Table 4). Preterm birth was associated with stunting and underweight at month 1 (aOR 2.7, 95%CI:1.2–6.3; aOR 5.4, 95%CI:1.7–16.1).

**Evolution of stunting among study population.** The overall HAZ means observed in the study population were -0.60 (95%CI: -0.7 to -0.5), -0.61 (95%CI: -0.5 to -0.4) and -0.87 (95% CI: -0.9 to -0.8) at months 1, 9 and 12, respectively. As shown in Fig 3, the mean HAZ remained negative in both groups over their first year of life. Among infants born with LBW, we observed an increase in HAZ from -1.68 (95%CI: -1.9 to -1.4) at month 1 to -0.97 (95%CI: -1.2 to -0.7) at month 9, followed by a decrease from month 9 of age to -1.29 (95%CI: -1.6 to -1.0) at 12 months of age. Among their counterpart born with normal weight, a slow decrease

**Table 3. Prevalence of stunting, wasting and underweight and univariate analysis of the risk factors.**

| | | N | Stunting | | Wasting | | Underweight | |
|---|---|---|---|---|---|---|---|---|
| | | | n (%) | p-value* | n (%) | p-value* | n (%) | p-value* |
| **Month 1** | | | | | | | | |
| | LBW | | | <0.0001 | | 0.1 | | <0.001 |
| | NO | 658 | 39 (5.9) | | 35 (5.3) | | 7 (1.0) | |
| | YES | 85 | 33 (38.8) | | 8 (9.4) | | 23 (27.1) | |
| | Preterm birth ** | | | 0.001 | | 0.3 | | <0.001 |
| | NO | 702 | 64 (9.1) | | 43 (6.1) | | 24 (3.4) | |
| | YES | 40 | 10 (25.0) | | 0 | | 8 (20.0) | |
| | Infant Sex | | | 0.1 | | 0.4 | | 0.8 |
| | Male | 380 | 43 (11.3) | | 20 (5.3) | | 16 (4.2) | |
| | Female | 363 | 29 (8.1) | | 23 (6.3) | | 14 (3.9) | |
| | Maternal Illiteracy | | | 0.4 | | 0.2 | | 0.8 |
| | NO | 607 | 61 (10.1) | | 32 (5.3) | | 25 (4.1) | |
| | YES | 136 | 11 (8.1) | | 11 (8.1) | | 5 (3.7) | |
| | Maternal age (year) | | | 0.06 | | 0.6 | | 0.5 |
| | <20 | 240 | 32 (13.3) | | 16 (6.7) | | 11 (4.6) | |
| | 20–35 | 442 | 36 (8.1) | | 26 (5.9) | | 18 (4.1) | |
| | >35 | 61 | 4 (6.6) | | 2 (3.2) | | 1 (1.6) | |
| **Month 9** | | | | | | | | |
| | LBW | | | 0.06 | | 0.004 | | 0.01 |
| | NO | 469 | 41 (8.7) | | 33 (7.0) | | 72 (13.2) | |
| | YES | 55 | 9 (16.4) | | 10 (18.2) | | 14 (25.5) | |
| | Preterm birth | | | 0.3 | | 0.1 | | 0.1 |
| | NO | 497 | 46 (9.3) | | 42 (8.5) | | 75 (3.7) | |
| | YES | 27 | 4 (14.8) | | 0 | | 1 (3.7) | |
| | Infant sex | | | 0.001 | | 0.6 | | 0.08 |
| | Male | 262 | 36 (13.7) | | 20 (7.6) | | 45 (17.1) | |
| | Female | 262 | 14 (5.3) | | 23 (8.8) | | 31 (11.8) | |
| | Maternal Illiteracy | | | 0.6 | | 0.006 | | 0.04 |
| | NO | 424 | 39 (9.2) | | 28 (6.6) | | 55 (13.0) | |
| | YES | 100 | 11 (11.0) | | 15 (15.0) | | 21 (21.0) | |
| | Maternal age (year) | | | 0.3 | | 0.4 | | 0.1 |
| | <20 | 156 | 19 (12.2) | | 9 (5.8) | | 24 (15.4) | |
| | 20–35 | 316 | 28 (8.9) | | 29 (9.2) | | 49 (15.5) | |
| | >35 | 52 | 3 (5.8) | | 5 (9.6) | | 3 (5.8) | |
| **Month 12** | | | | | | | | |
| | LBW | | | 0.004 | | 0.1 | | 0.001 |
| | NO | 470 | 71 (15.1) | | 30 (6.4) | | 82 (17.5) | |
| | YES | 60 | 18 (30.8) | | 7 (11.7) | | 28 (46.7) | |
| | Preterm birth | | | 0.5 | | 0.1 | | 0.4 |
| | NO | 498 | 85 (17.1) | | 37 (7.4) | | 105 (21.1) | |
| | YES | 32 | 4 (12.5) | | 0 | | 5 (15.6) | |
| | Infant Sex** | | | <0.001 | | 0.8 | | 0.002 |
| | Male | 266 | 67 (25.2) | | 17 (6.7) | | 70 (26.3) | |
| | Female | 263 | 22 (8.4) | | 19 (7.2) | | 41 (15.6) | |
| | Maternal Illiteracy** | | | 0.1 | | 0.02 | | 0.01 |
| | NO | 421 | 66 (15.7) | | 24 (5.7) | | 78 (18.5) | |

*(Continued)*

**Table 3.** (Continued)

| | N | Stunting | | Wasting | | Underweight | |
|---|---|---|---|---|---|---|---|
| | | n (%) | *p-value*[*] | n (%) | *p-value*[*] | n (%) | *p-value*[*] |
| YES | 108 | 23 (21.3) | | 13 (12.0) | | 32 (30.5) | |
| Maternal age (year) ** | | | 0.6 | | 0.5 | | 0.5 |
| <20 | 158 | 30 (19.0) | | 14 (8.7) | | 32 (20.2) | |
| 20–35 | 318 | 51 (16.0) | | 20 (6.3) | | 70 (22.3) | |
| >35 | 53 | 8 (15.1) | | 3 (5.7) | | 8 (15.1) | |

SD: standard deviation, N/n:total population, LBW: Low Birth Weight.

[*] chi square test.

[**] 1 missing data.

of the HAZ curve was observed from month 1 of life from -0.46 (95%CI: -0.5 to -0.4) to -0.57 (95%CI: -0.7 to -0.5) at 9 months and to -0.81 (95%CI: -0.9 to -0.7) at 12 months.

## Mortality during the first year of life

A total of 25 deaths were recorded among the 907 infants included in the study, giving an infant mortality rate of 28 deaths per 1000 live births (95% CI: 17–30). With regards to the

**Table 4. Assessment of the impact of birth weight and term of birth on stunting, wasting and underweight in infancy.**

| | | | Stunting | | Wasting | | Underweight | |
|---|---|---|---|---|---|---|---|---|
| | | | OR [95%CI] | aOR[95%CI] | OR [95%CI] | aOR[95%CI] | OR [95%CI] | aOR[95%CI] |
| **Month 1** | | | | | | | | |
| | LBW | | | | | | | |
| | | No | 1 | 1 | 1 | 1 | 1 | 1 |
| | | Yes | 10.4 [5.6–17.8] | 10.3 [5.9–17.9] | 2.1 [0.9–4.5] | 2.2 [0.9–4.7] | 34.5 [14.2–83.6] | 33.1 [12.2–83.2] |
| | Preterm birth | | | | | | | |
| | | No | 1 | 1 | 1 | 1 | 1 | 1 |
| | | Yes | 3.6 [1.7–7.5] | 2.7 [1.2–6.3] | 0.4 [0.5–2.8] | 0.3 [0.0–2.3] | 7.0 [3.0–17.0] | 5.3 [1.4–16.0] |
| **Month 9** | | | | | | | | |
| | LBW | | | | | | | |
| | | No | 1 | 1 | - | - | 1 | 1 |
| | | Yes | 2.0 [0.9–4.5] | 1.9 [0.9–4.3] | | - | 2.2 [1.2–4.3] | 2.2 [1.1–4.4] |
| | Preterm birth | | | | | | | |
| | | No | 1 | 1 | - | - | 1 | 1 |
| | | Yes | 1.7 [0.6–5.1] | 1.8 [0.6–5.7] | | - | 0.2 [0.0–1.5] | 0.2 [0.0–1.4] |
| **Month 12** | | | | | | | | |
| | LBW | | | | | | | |
| | | No | 1 | 1 | - | - | 1 | 1 |
| | | Yes | 2.4 [1.3–4.4] | 2.6 [1.4–4.9] | | - | 4.1 [2.3–7.2] | 4.5 [2.5–7.2] |
| | Preterm birth | | | | | | | |
| | | No | 1 | 1 | - | - | 1 | 1 |
| | | Yes | 0.7 [0.2–2.0] | 0.6 [0.2–2.2] | | - | 0.7 [0.3–1.8] | 0.5 [0.1–1.5] |

OR: Odds Ratio, aOR: adjusted Odds Ratio; 95%CI: 95% confidence interval, LBW: Low Birth Weight.

Adjusted for infant sex, maternal age and maternal literacy; there was no wasting at 9 and 12 months among preterm infants, so no estimates of OR could be made for these timepoints for wasting.

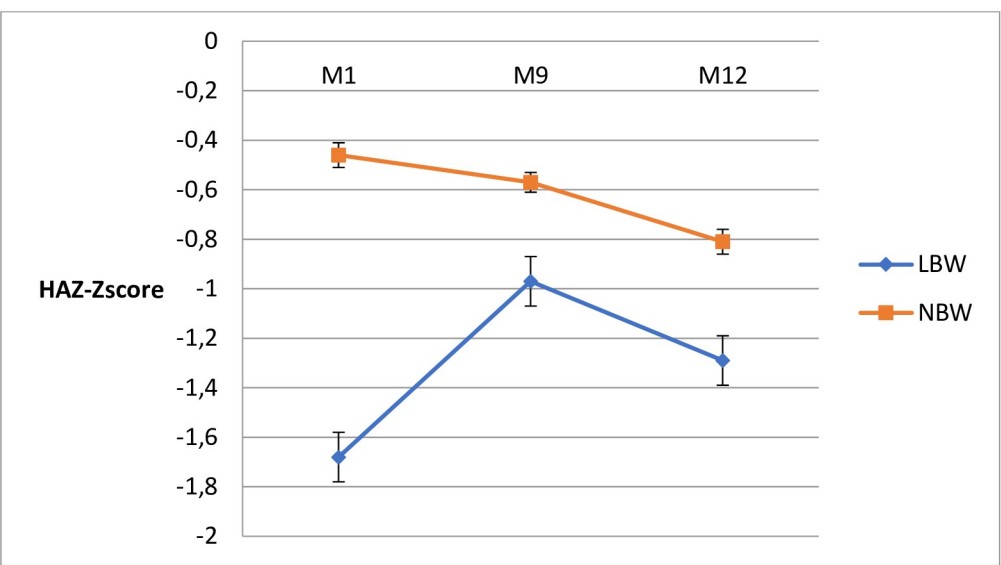

**Fig 3. Mean (SE) height-for-age Z-score evolution from 1 to 12 months by birth weight.** Abbreviations: M1: month one, M9: month nine; M12: month twelve; LBW: low birth weight; NBW: normal birth weight.

study groups, the infant mortality in the first year of life was 18 deaths per 1000 (95% CI: 8.8–27) live births (14 deaths) for infants with NBW and 96 deaths per 1000 (95% CI: 91.3–100.7) for infant with LBW, respectively. The survival curve of infants with LBW compared with those born with NBW demonstrates a statically significant higher number of deaths over the whole observation period ($p < 0.002$) (Fig 4).

Moreover, comparing both study groups, the crude hazard ratio shows a 6.4 (95% CI: 1.95–20.98) increased hazard of death among children with LBW compared to those with NBW. Adjusted for preterm birth, infant sex and mother age and literacy, infants with LBW had a 4.5-fold higher hazard to die than infants with NBW (Table 5).

## Discussion

Our findings show a high prevalence of low birth weight in the study area in Gabon with 13% babies born too small. There was a significant weight gain in the LBW infants but that was not enough to reach the mean weight of infants born with normal weight. Malnutrition indicators that were stunting, wasting and underweight were significantly more frequent in the LBW infants' group, and so was the mortality.

Low birthweight is known to be associated with poor postnatal growth, particularly during the first year of life [18, 28]. However, the impact of LBW varies between respective settings, which makes it important to obtain local data. The impact of LBW on nutritional deficits and survival was never assessed in Gabon and few data exist for the Central African region. Weight, nutritional status, and mortality are among the best indicators to assess an infant's wellness in his first year of life.

The observed higher weight gain among LBW infants during the first year of life is in line with a previous report indicating a difference in mean weight gain up to 8 months of age [29]. Sridhar et al. [30] and Borah et al. [31] also reported higher weight gains in LBW infants compared to their normal birth weight counterpart. This can potentially be explained by the particular attention that mothers and health caregivers may be providing to infants born too small. Indeed, in the maternal and infant health services there is a screening of malnutrition and

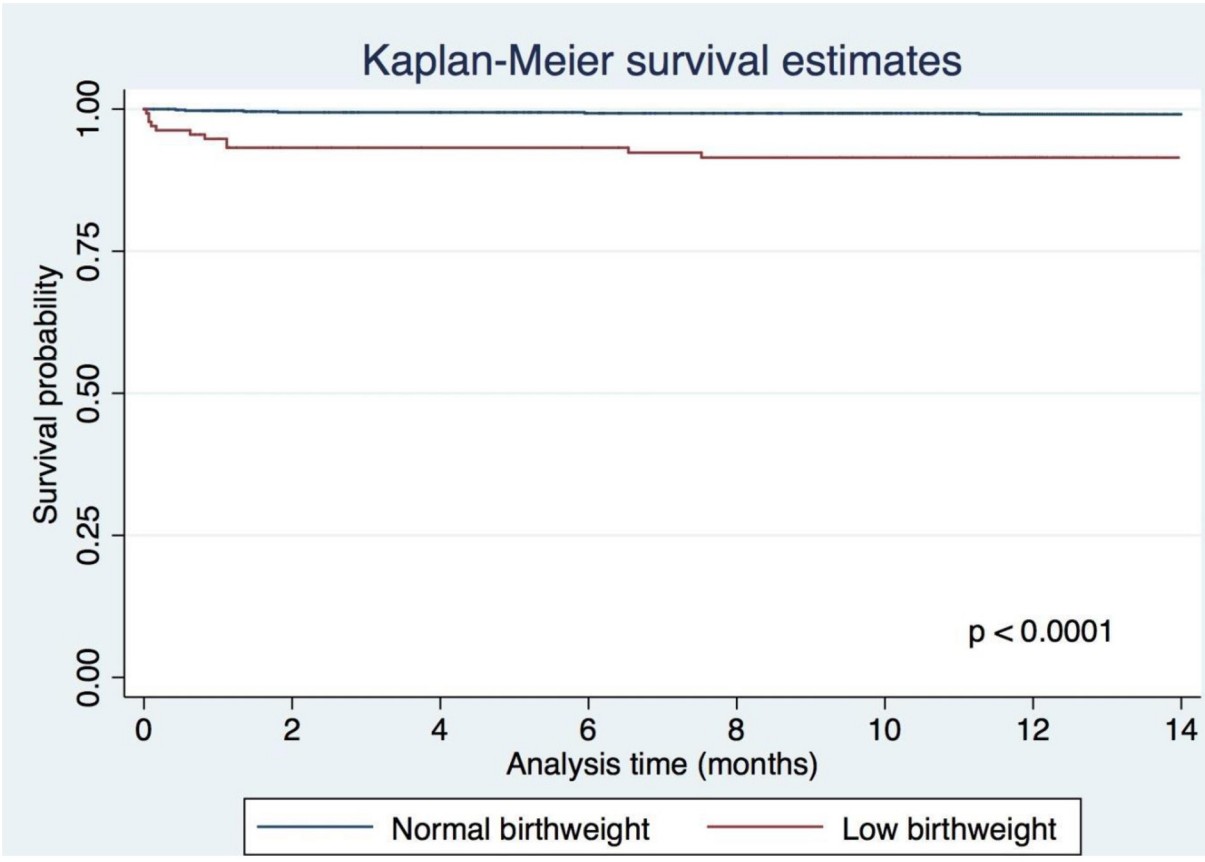

**Fig 4. Comparison of survival stratified by birth weight.**

nutritional advice and practical exercises are provided to mothers with too small infants. Nonetheless, it was observed that despite that particular attention and the observed significant weight gain among LBW infants, there remains a difference in mean weight with the normal weight born infants. This finding corroborates with the results from a study in Burkina Faso in which WAZ was used as an indicator of malnutrition during the first year of life, and it was also observed that LBW children's growth curve remained below that of normal birthweight children [29].

The results show strong evidence that LBW was an independent risk factor for stunting, wasting and underweight while that was not the case for preterm birth. This suggests that the small for gestational age component of LBW which is in utero growth restriction may be more

**Table 5. Effect of low birth weight on infant mortality.**

|  | Univariable model | | Multivariable model** | |
|---|---|---|---|---|
|  | **Hazard ratio (95% CI)** | **p-value*** | **Hazard ratio (95% CI)** | **p-value*** |
| **NBW** | 1 | | 1 | |
| **LBW** | 6.46 (1.97–21.18) | 0,002 | 4.55 (1.19–17.30) | 0.026 |

N = 813 for both the univariable and the multivariable models; LBW: low Birth Weight, NBW: Normal Birth Weight.

* Wald test.

**Adjusted for preterm birth, infant sex, maternal age and maternal literacy.

important in explaining the observed postnatal growth retardation rather than the being born too soon component in our study population. Actually, in the absence of congenital malformations or chromosomal abnormalities foetal size could be the consequence of two distinct processes: constitutional smallness or pathological growth restriction and distinguishing one process from the other is challenging [32]. In our study, infants have been categorized according to their birth weight without checking if a growth restriction really occurred. It is known that growth restricted foetuses are small because of some underlying pathological conditions including uteroplacental dysfunction, hypertensive disorders or illicit and toxic substances during pregnancy such as smoking. The observed important weight gain in the LBW infants' group may suggest that the constitutional smallness process may be negligible in this population. The implication of the LBW mechanism involved in these infants' adverse growth and nutrition outcomes would therefore be more explicit if more variables reflecting the potential underlying pathological conditions but also the socioeconomic status of the parents and family size, infant breastfeeding and other nutritional interventions addressed to infants were recorded and controlled for.

The observed overall mortality rate of 28 deaths per 1000 live births was significantly below the 81 deaths per 1000 live births reported by UNICEF in 2016 for the Central African region [2]. This could be explained by the clinical trial context as the MIPPAD study that recruited the participants was a funded clinical trial providing basic health care, and HIV-positive pregnant women were systematically excluded. Most deaths occurred before the second month of life and that is similar to what was reported by previous studies from Burkina Faso and Ghana [20, 29]. There was a strong evidence of an association between LBW and infant mortality in our study, in line with reports from other settings [33, 34].

There were a lot of infants from baseline and month 1 that missed the months 9 and 12 visits and this could be a huge limitation. The high loss to follow-up here can be due to the high mobility of the population in Gabon and it can be interpreted that around the age when infants start sitting and crawling, they are taken into trips with other family members. That was addressed by conducting sensitivity analyses that showed that at 9-month visit there were more infants from the LBW group missing compared to the normal birth weight group. That suggests that the observed effect was certainly misclassified and underestimated.

Among the strengths of this study are the prospective design, the high-quality standardized data collection and follow-up guaranteed by the clinical trial context as well as the high coverage of standard antenatal and postnatal care financially supported that could limit the accessibility and availability of care barriers frequently described in resource-limited settings. The systematic exclusion of HIV positive pregnant women is limiting the generalization of the results of this study and residual confounding may not be completely ruled out as the randomization in the MiPPAD trial was based on the intervention which was not taken into account in this analysis. The fixed time points for assessment that were at months 1, 9 and 12 and the relatively short period of follow-up on one year could also be limiting as important outcomes particularly later in life are not captured. It may therefore be advised for future studies to have monthly visits and a longer follow-up period to account for time-dependent variations of the outcomes.

## Conclusion

Our findings that LBW is highly frequent among babies born from HIV-negative pregnant women from our study area in Gabon and its association with infants' restricted growth, nutritional deficits and mortality, advocate for the strengthening of health interventions targeting the prevention of low birth weight in women. It may be too late to prevent the adverse

nutrition and mortality outcomes after birth in the babies born too small but an emphasis on nutritional interventions must be provided for them. Studies designed to distinguish the small for age and intrauterine growth restriction components of LBW are recommended to further understand the mechanisms involved and develop interventions accordingly.

## Supporting information

**S1 Table. Sensitivity analysis.**
(DOCX)

## Acknowledgments

We are thankful to the MiPPAD Consortium and the staff at CERMEL, at the Centre Médical de Fougamou and at the Centre de Recherches Médicales de la Ngounié in Fougamou, at the Albert Schweitzer hospital and at the Centre Hospitalier Régional Georges Rawiri in Lambaréné. We are also grateful to the women and families that participated in the MiPPAD study in the different study sites in Gabon.

## Author Contributions

**Conceptualization:** Rella Zoleko-Manego, Raquel González, Clara Menendez, Michael Ramharter, Ghyslain Mombo-Ngoma.

**Data curation:** Rella Zoleko-Manego.

**Formal analysis:** Rella Zoleko-Manego, Johannes Mischlinger, Jean Claude Dejon-Agobé, Clara Menendez, Michael Ramharter.

**Funding acquisition:** Peter G. Kremsner, Raquel González, Clara Menendez, Michael Ramharter, Ghyslain Mombo-Ngoma.

**Investigation:** Rella Zoleko-Manego, Arti Basra, J. Rodolphe Mackanga, Daisy Akerey Diop, Ghyslain Mombo-Ngoma.

**Methodology:** Rella Zoleko-Manego, Clara Menendez, Ghyslain Mombo-Ngoma.

**Project administration:** Rella Zoleko-Manego, Ayola Akim Adegnika, Selidji T. Agnandji, Bertrand Lell, Peter G. Kremsner, Raquel González, Clara Menendez, Michael Ramharter.

**Resources:** Ayola Akim Adegnika, Selidji T. Agnandji, Bertrand Lell, Peter G. Kremsner, Pierre Blaise Matsiegui, Ghyslain Mombo-Ngoma.

**Software:** Ayola Akim Adegnika, Selidji T. Agnandji, Bertrand Lell.

**Supervision:** Pierre Blaise Matsiegui, Clara Menendez, Michael Ramharter, Ghyslain Mombo-Ngoma.

**Validation:** Rella Zoleko-Manego, Ghyslain Mombo-Ngoma.

**Visualization:** Rella Zoleko-Manego.

**Writing – original draft:** Rella Zoleko-Manego, Johannes Mischlinger, Jean Claude Dejon-Agobé, Ghyslain Mombo-Ngoma.

**Writing – review & editing:** Rella Zoleko-Manego, Johannes Mischlinger, Jean Claude Dejon-Agobé, Arti Basra, J. Rodolphe Mackanga, Daisy Akerey Diop, Ayola Akim Adegnika, Selidji T. Agnandji, Bertrand Lell, Peter G. Kremsner, Pierre Blaise Matsiegui, Raquel González, Clara Menendez, Michael Ramharter, Ghyslain Mombo-Ngoma.

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
