## [Decision Letter · Decision Letter 0]

17 Sep 2020

PONE-D-20-19873

Birth weight, growth, nutritional status and mortality of infants from Lambaréné and Fougamou in Gabon in their first year of life

PLOS ONE

Dear Dr. Mombo-Ngoma,

Thank you for submitting your manuscript to PLOS ONE. After careful consideration, we feel that it has merit but does not fully meet PLOS ONE’s publication criteria as it currently stands. Therefore, we invite you to submit a revised version of the manuscript that addresses the points raised during the review process.

During the review please address the following major issues.

1. Please explore the the issue of loss to follow up bias by statistically comparing individuals retained and dropped out of the study at different stages of the study. 

2. Please provide a clear and objective criteria how the potential confounders were selected for statistical adjustment. 

We look forward to receiving your revised manuscript.

Kind regards,

Samson Gebremedhin, PhD

Academic Editor

PLOS ONE

Journal Requirements:

'I have read the journal's policy and the authors of this manuscript have the following competing interests: CM is a member of the Editorial Board of PLOS Medicine. The authors have declared that no other competing interests exist'

a. Please confirm that this does not alter your adherence to all PLOS ONE policies on sharing data and materials, by including the following statement: "This does not alter our adherence to  PLOS ONE policies on sharing data and materials.” (as detailed online in our guide for authors http://journals.plos.org/plosone/s/competing-interests).  If there are restrictions on sharing of data and/or materials, please state these.

Please note that we cannot proceed with consideration of your article until this information has been declared.

Reviewers' comments:

Reviewer's Responses to Questions

**Comments to the Author**

1. Is the manuscript technically sound, and do the data support the conclusions?

Reviewer #1: Partly

Reviewer #2: Yes

2. Has the statistical analysis been performed appropriately and rigorously? 

Reviewer #1: No

Reviewer #2: Yes

3. Have the authors made all data underlying the findings in their manuscript fully available?

Reviewer #1: No

Reviewer #2: No

4. Is the manuscript presented in an intelligible fashion and written in standard English?

Reviewer #1: No

Reviewer #2: Yes

5. Review Comments to the Author

Reviewer #1: This manuscript is on an important topic, but the writing is somewhat confusing and needs a close grammar proofread; not all of the numbers in the text match the tables; and I have certain concerns about the data and analyses as presented.

Major concerns:

My 2 major concerns are about the amount of missing data and the selected confounders. The most concerning issue for me is that loss to follow up is a huge limitation. The study lost either 25 or about 45% of the sample, depending on how you choose the denominator (one month vs. baseline). This issue is not addressed anywhere in the limitations or in the manuscript. I think the authors need to present some sensitivity analyses and say something about how they differed than the study participants who were retained, and how those differences might influence the study's findings. It's also not clear which children are in the KM and hazard analyses--have the authors censored the LTFs at last seen date, or just ignored them in that analysis?

My other major concern is that the authors have some specific confounders they controlled for, but why these specific ones were chosen are mysterious. There are other factors also possibly relevant--breast feeding history, complementary feeding history, family wealth, number of other kids in household, availability of & use of antenatal care--so what was the rationale for including these specific confounders and nothing else?

Minor concerns

Line 68-71: this sentence is hard to understand. I'd break it into 2 because you seem to have 2 concepts here.

Line 80: "Once established, stunting and its effects typically become permanent. Stunting is an appropriate indicator for chronic malnutrition"...There's some controversy about this. Some studies have shown trajectories in which children recover and others in which they don't. Early intervention is believed to help with this. You need some citations for your contention here.

Line 82:"Underweight is most commonly used as nutritional indicator due to difficulties..." This statement again needs some citation. DHS,. MICS, and a lot of other national programs use stunting and wasting. Saying that underweight is most commonly used needs to be backed up with some evidence.

Line 133: "For those with no weight available at birth but in the first week of life in case of home deliveries weight was estimated using a regression model". This needs some explanation. How do you estimate weight with regression?

Table 1: Preterm birth totals don't add to 907 or 115. I suggest that the authors present "n"s when they don't match the stated denominator.

Line 189: Authors haven't defined a weight gain ratio.

Line 200: if the authors are presenting decimal places in the table, it makes sense to match that in the text.

Line 208: "There were more significantly observed underweight LBW infants compared to their normal weight counterpart". I did not understand the sentence. what does it mean to be significantly observed underweight?

Lines 200-212: I found it challenging to read and follow this paragraph. Maybe the authors could restructure this so that they talk about each endpoint separately? So, first stunting, and what's associated. Then wasting, and what's associated with that, etc.

Lines 221-222: These numbers should all match what is in table 4. I found this paragraph also very confusing. LBW was associated with higher stunting at months 1 AND month 12, but not in between . These are all one model, correct? So preterm birth is associated with stunting independent of LBW ?

Discussion: the English here needs some boosting also.

Lines 263-265: I didn't understand what it means to have a nutritional gain if the prevalence of stunting and underweight increased and prevalence of wasting stayed the same. Perhaps reword this? Do you mean that mean weight increased for all children?

Lines 272-274: "We hypothesize that this catching up in weight gain is at least in part due to the particular attention caused by mothers and the health care system allowing children born with LBW to acquire a normal weight" What is this extra attention? Can you be more specific about Gabon's practices in the health care system to address these nutritional challenges in LBW children?

Lines 293-296: I don;t understand this paragraph. LBW is an independently associated with stunting at 12 months in this setting--but what does that have to do with weight deficit since birth? These 2 things are probably true, but the authors haven't really connected them coherently here.

Lines 303-304: "This indicates that 302 health interventions need to focus primarily on the two first months of life, particularly for newborns with LBW." I'm not sure this is a reasonable suggestion. What do the authors mean by "primarily"? Even among those who survive, a high proportion have poor nutrition outcomes. So I'm not sure it makes sense to say that that health care programs should put most focus on the first 2 months. Obviously something needs to be done but the children who survive also need health care support. Also, the authors don;t mention what proportion of deaths occur as neonates. Health interventions often don't help to move the needle on neonatal death. What might help with these early deaths is antenatal care, facility delivery in a well-stocked facility, etc. Other health service interventions may be of more benefit in children after the first few weeks of life.

Reviewer #2: The manuscript presented an interesting data on the significance of LWB for survival and nutritional status of infants in Gabon. While the findings of the analysis are not new, they may help to consolidate the existing knowledge on the topic. In general, the manuscript is also well written.

I recommend the authors accommodate the following comments.

Exhaustive list of potential confounders along with approach used for selecting the variables for multivariable adjustment should be provided. For example, it is not clear how the authors decided to adjust the multivariable models for infant sex, maternal age and maternal literacy.

The adequacy of the sample size for comparing mortality between LBW and normal birthweight infants should be evaluated through post-hoc power calculation.

The possibility of loss to follow up bias should be explored by comparing the characteristics of the study participants retained and dropped out of the study.

The basic characteristics of exposed and non-exposed subjects should be described and statistically compared at the beginning of the Results section.

6. PLOS authors have the option to publish the peer review history of their article (what does this mean?). If published, this will include your full peer review and any attached files.

Reviewer #1: No

Reviewer #2: No

---

## [Author Response · Author response to Decision Letter 0]

8 Nov 2020

Reviewer #1:

This manuscript is on an important topic, but the writing is somewhat confusing and needs a close grammar proofread; not all of the numbers in the text match the tables; and I have certain concerns about the data and analyses as presented.

Reply: We thank the reviewer for the thorough assessment of our manuscript and the queries raised. This has tremendously been helpful in improving the quality of the manuscript. Please, we have performed a grammar proofread and corrected the mismatches in the tables.

Major concerns:

My 2 major concerns are about the amount of missing data and the selected confounders. The most concerning issue for me is that loss to follow up is a huge limitation. The study lost either 25 or about 45% of the sample, depending on how you choose the denominator (one month vs. baseline). This issue is not addressed anywhere in the limitations or in the manuscript. I think the authors need to present some sensitivity analyses and say something about how they differed than the study participants who were retained, and how those differences might influence the study's findings. 

Reply: This is correct that there were a lot of losses to follow-up from 9 months that can be explained by the high mobility of the population in Gabon. This limitation has been addressed by conducting sensitivity analyses as advised by the reviewer. The findings of the sensitivity analyses are shown in the supplementary table added. Mention of the sensitivity analyses have been added in the methods, results and discussion sections.

It's also not clear which children are in the KM and hazard analyses--have the authors censored the LTFs at last seen date, or just ignored them in that analysis?

Reply: 907 infants were included in the KM hazard analysis. LTFs were included in the analysis and censored at the last contact date. 

My other major concern is that the authors have some specific confounders they controlled for, but why these specific ones were chosen are mysterious. There are other factors also possibly relevant--breast feeding history, complementary feeding history, family wealth, number of other kids in household, availability of & use of antenatal care--so what was the rationale for including these specific confounders and nothing else?

Reply: Some factors including mother’s age and literacy, infant’s sex and prematurity, breast feeding history, complementary feeding history, family wealth, number of other kids in household, availability of & use of antenatal care are recognised as risk factors for malnutrition. The main study was conducted to evaluate intermittent preventive treatment of malaria during pregnancy, for which some of those variables were recorded and used in our analysis. The main study was designed as a randomized clinical trial with the assumption that confounders would be evenly distributed between the treatment arms, they were therefore not systematically recorded. We have mentioned that limitation in the manuscript (line 306-307). It can be believed that the socioeconomic factors have minimal effect as in the context of the clinical trial as standard care was provided to all participants and their offspring.

Minor concerns

Line 68-71: this sentence is hard to understand. I'd break it into 2 because you seem to have 2 concepts here.

Reply: Sentence was broken into two sentences as advised

Line 80: "Once established, stunting and its effects typically become permanent. Stunting is an appropriate indicator for chronic malnutrition"...There's some controversy about this. Some studies have shown trajectories in which children recover and others in which they don't. Early intervention is believed to help with this. You need some citations for your contention here.

Reply: Thank you for this suggestion. We added a citation (Dewey and Begum, 2011) to support the statement

Line 82:"Underweight is most commonly used as nutritional indicator due to difficulties..." This statement again needs some citation. DHS, MICS, and a lot of other national programs use stunting and wasting. Saying that underweight is most commonly used needs to be backed up with some evidence.

Reply: We added citation (Prendergast and Humphrey, 2014) to support the statement. 

Line 133: "For those with no weight available at birth but in the first week of life in case of home deliveries weight was estimated using a regression model". This needs some explanation. How do you estimate weight with regression?

Action: There is a linear regression model described by Greenwood and colleagues (1992) adjusting the birthweight measured after the day of birth. On average, infants’ weight 6.3 grams less each day after birth. In the analysis the weight at birth was estimated using this coefficient for those children weighed after day 0. The fitted values by the linear model are very similar to the values estimated by Lowess regression.

Below, the estimations from the main trial as published in Plos Medicine (Gonzales et al. 2014)

Table 1: Preterm birth totals don't add to 907 or 115. I suggest that the authors present "n"s when they don't match the stated denominator.

Reply: The reviewer is correct on that point, we have corrected the table with “n”s for each variable and since the total population at baseline was 907, we have indicated with “*” the variables with missing data. 

Line 189: Authors haven't defined a weight gain ratio.

Reply: This is correct, it has now been defined in the text. Weight gain ratio is the ratio of the weight of infants at the visit divided by the weight of the infants at the previous visit. It has been added in the manuscript line 155-157.

Line 200: if the authors are presenting decimal places in the table, it makes sense to match that in the text.

Reply: We rounded in order to make the result easier to read but reviewer’s comment is relevant so we have shown the decimal places in the text as well.

Line 208: "There were more significantly observed underweight LBW infants compared to their normal weight counterpart". I did not understand the sentence. what does it mean to be significantly observed underweight?

Reply: The sentence was reformulated. “There were significantly more underweight observed among infants born with low weight compared to their normal birth weight counterpart” (line 217-219)

Lines 200-212: I found it challenging to read and follow this paragraph. Maybe the authors could restructure this so that they talk about each endpoint separately? So, first stunting, and what's associated. Then wasting, and what's associated with that, etc.

Reply: Text was restructured

Lines 221-222: These numbers should all match what is in table 4. I found this paragraph also very confusing. LBW was associated with higher stunting at months 1 AND month 12, but not in between. These are all one model, correct? So preterm birth is associated with stunting independent of LBW?

Reply: Numbers were corrected in the text to match what is in table 4. That’s the same model, and it showed actually stronger evidence of an association of LBW and stunting at month 1 and month 12. For month 9, there was obviously a trend of an association too as the estimates of OR was 2.0, however the 96%CI crossed 0. That weak evidence at 9 months is not yet fully understood but we think that it reflects what is shown in Figure 3 as there is an important weight gain before 9 months.

Discussion: the English here needs some boosting also.

Reply: The English was reviewed and the section was rewritten to address the language issue

Lines 263-265: I didn't understand what it means to have a nutritional gain if the prevalence of stunting and underweight increased and prevalence of wasting stayed the same. Perhaps reword this? Do you mean that mean weight increased for all children?

Reply: This was actually a mistake that has been corrected, thank you. We meant weight gain instead.

Lines 272-274: "We hypothesize that this catching up in weight gain is at least in part due to the particular attention caused by mothers and the health care system allowing children born with LBW to acquire a normal weight" What is this extra attention? Can you be more specific about Gabon's practices in the health care system to address these nutritional challenges in LBW children?

Reply: We mean that the important weight gain observed in the LBW infants’ group may be explained by the particular attention given to that group and the subsequent restoration of growth that restricted in utero. 

Lines 293-296: I don’t understand this paragraph. LBW is an independently associated with stunting at 12 months in this setting--but what does that have to do with weight deficit since birth? These 2 things are probably true, but the authors haven't really connected them coherently here.

Reply: The sentence was rephrased to highlight the potential mechanisms of LBW involved, see action above.

Lines 303-304: "This indicates that 302 health interventions need to focus primarily on the two first months of life, particularly for newborns with LBW." I'm not sure this is a reasonable suggestion. What do the authors mean by "primarily"? Even among those who survive, a high proportion have poor nutrition outcomes. So, I'm not sure it makes sense to say that that health care programs should put most focus on the first 2 months. Obviously, something needs to be done but the children who survive also need health care support. Also, the authors don’t mention what proportion of deaths occur as neonates. Health interventions often don't help to move the needle on neonatal death. What might help with these early deaths is antenatal care, facility delivery in a well-stocked facility, etc. Other health service interventions may be of more benefit in children after the first few weeks of life.

Reply: We have reformulated our recommendations and it is indeed better to promote interventions targeting pregnant women to prevent babies to be born too small.

Reviewer #2:

The manuscript presented an interesting data on the significance of LWB for survival and nutritional status of infants in Gabon. While the findings of the analysis are not new, they may help to consolidate the existing knowledge on the topic. In general, the manuscript is also well written. I recommend the authors accommodate the following comments:

-Exhaustive list of potential confounders along with approach used for selecting the variables for multivariable adjustment should be provided. For example, it is not clear how the authors decided to adjust the multivariable models for infant sex, maternal age and maternal literacy.

Reply: Some factors including mother’s age and literacy, infant’s sex and prematurity, breast feeding history, complementary feeding history, family wealth, number of other kids in household, availability of & use of antenatal care are recognised as risk factors for malnutrition. The main study was conducted to evaluate intermittent preventive treatment of malaria during pregnancy, for which some of those variables were recorded and used in our analysis. The main study was designed as a randomized clinical trial with the assumption that confounders would be evenly distributed between the treatment arms, they were therefore not systematically recorded. We have mentioned that limitation in the manuscript (line 306-307). It can be believed that the socioeconomic factors have minimal effect as in the context of the clinical trial as standard care was provided to all participants and their offspring.

-The adequacy of the sample size for comparing mortality between LBW and normal birthweight infants should be evaluated through post-hoc power calculation.

Reply: we calculated post-hoc power for the comparison of mortality between LBW and normal birth weight and we obtained 95 %.

-The possibility of loss to follow up bias should be explored by comparing the characteristics of the study participants retained and dropped out of the study.

Reply: This is correct, we have now addressed that limitation by conducting sensitivity analyses that showed that the baseline characteristics of the retained participants were similar to that of the whole population including the lost to follow up. This is reported in the supplementary table.

-The basic characteristics of exposed and non-exposed subjects should be described and statistically compared at the beginning of the Results section.

Reply: We included exposed and non-exposed subjects in baseline characteristics table.

---

## [Decision Letter · Decision Letter 1]

7 Dec 2020

PONE-D-20-19873R1

Birth weight, growth, nutritional status and mortality of infants from Lambaréné and Fougamou in Gabon in their first year of life

PLOS ONE

Dear Dr. Mombo-Ngoma,

Thank you for submitting your manuscript to PLOS ONE. After careful consideration, we feel that it has merit but does not fully meet PLOS ONE’s publication criteria as it currently stands. Therefore, we invite you to submit a revised version of the manuscript that addresses the points raised during the review process. Specially I recommend you to address issues raised on the sensitivity analysis and the approach used for selecting variables for the statistical adjustment. 

We look forward to receiving your revised manuscript.

Kind regards,

Samson Gebremedhin, PhD

Academic Editor

PLOS ONE

Reviewers' comments:

Reviewer's Responses to Questions

**Comments to the Author**

1. If the authors have adequately addressed your comments raised in a previous round of review and you feel that this manuscript is now acceptable for publication, you may indicate that here to bypass the “Comments to the Author” section, enter your conflict of interest statement in the “Confidential to Editor” section, and submit your "Accept" recommendation.

Reviewer #1: (No Response)

Reviewer #2: (No Response)

2. Is the manuscript technically sound, and do the data support the conclusions?

Reviewer #1: Partly

Reviewer #2: Yes

3. Has the statistical analysis been performed appropriately and rigorously? 

Reviewer #1: No

Reviewer #2: Yes

4. Have the authors made all data underlying the findings in their manuscript fully available?

Reviewer #1: Yes

Reviewer #2: (No Response)

5. Is the manuscript presented in an intelligible fashion and written in standard English?

Reviewer #1: Yes

Reviewer #2: Yes

6. Review Comments to the Author

Reviewer #1: I thank the authors for their efforts to address the comments that the other reviewer and I made. I appreciate their time. However, I still have one large concern and a few minor changes and suggestions.

MAIN large reservation:

I don’t think that the authors entirely understood my request to do a sensitivity analysis of those who were lost to follow up, or if they did, it’s not reflected in the supplemental table. In lines 317-323 “sensitivity analyses that demonstrated that the baseline characteristics of the retained children were not different from the entire population at baseline [**assuming that those dropped out were similar to those retained with regard to these characteristics**].” I have bracketed the concerning portion because in my view, that assumption renders the authors’ sensitivity analysis invalid.

What I am thinking of here is a sensitivity analysis in its simplest form as follows: those who were LTFU are one category and those who were never LTFU or missing are another. Then compare those 2 categories on things like LBW, malnutrition, preterm birth, infant sex, etc., What I see in the Supplementary table are descriptions of those who were retained at different time points, with no p values or comparisons. Just based on the numbers I see in the table, at 9 mo, the authors have 60/115 LBW kids (~52%) vs 515/792 normal birthweight kids (~65%). Which could very well be a statistically significant difference, but we don't know. The mean birthweight in your LBW children at 12 months was somewhat higher than in the LBW children at baseline. My point is that the children that were lost may be different in meaningful ways than the ones who were retained. My suggestion is to figure out whether there were differences, present p values in the supplement table to reassure the readers, and if there are differences, mention in the discussion whether you think they’d impact the results and how. It may be that you suspect that if everyone were there, you’d get even stronger associations.

Minor comments and proofreading:

Line 35, don’t need comma after “both”

Line 40 : included between 2009 and 2012 makes it sound like these are “n”s . Suggest you move the word “included” to after 2012.

Line 48: It could be the same number of infants that were LBW, it’s just that you’ve lost more during follow up. What are the actual ns? Is it same kids?

Line 55: ratio, not ration.

Body:

line 77 “”stunting is reported to be a strong marker of healthy growth”. It’s more like unhealthy growth, no?

line 86” While stunting wasting and underweight…low birth weight…” maybe make that the first sentence of the LBW paragraph

line 139: “like describe elsewhere”. This is not great grammar and not especially useful. I’d suggest you say something like “calculated using standard formula” and cite maybe DHS statistics or WHO web page that talks about how to calculate,

line 139

Height-for-age Z-score (HAZ), weight-for-age Z-score (WAZ) and weight-for-height Z-score (WHZ) were used to define respectively stunting, underweight and wasting. HAZ <-2 standard deviations (SD), WAZ <-2 141SD and WHZ <-2 SD.

Suggest you reword to make complete sentences:

Ex: Height-for-age Z-score (HAZ, <-2 standard deviations below an international reference mean), weight-for-age Z-score (WAZ <-2 SD), and weight-for-height Z-score (WHZ <-2 SD) were used to define respectively stunting, underweight and wasting.

144: data”were” . Data is a plural

Line 148: on comparing baseline characteristics of infants attending each visit to explore whether there was any alteration that could be imputed to the lost to follow-up.

Sentence is confusing. try instead “to explore whether any differences existed that could be associated with loss to follow up”

Table 4: I’ve forgotten why Wasting is only assessed at month 1. I’d suggest the authors add a footnote to the table about that, because the authors obviously calculated prevalence of wasting at all times.

Reviewer #2: Most of my concerns have been addressed. However, it remains vague how the variables infant sex, maternal age and maternal literacy were selected for adjustments. What possible control variables did the authors considered at the beginning of the analysis? How did the end up in the three variables? This must be clearly described in the manuscript.

7. PLOS authors have the option to publish the peer review history of their article (what does this mean?). If published, this will include your full peer review and any attached files.

Reviewer #1: No

Reviewer #2: No

---

## [Author Response · Author response to Decision Letter 1]

21 Jan 2021

Rebuttal point-by-point response tot he reviewers

Review Comments to the Author

Reviewer #1: I thank the authors for their efforts to address the comments that the other reviewer and I made. I appreciate their time. However, I still have one large concern and a few minor changes and suggestions.

MAIN large reservation:

I don’t think that the authors entirely understood my request to do a sensitivity analysis of those who were lost to follow up, or if they did, it’s not reflected in the supplemental table. In lines 317-323 “sensitivity analyses that demonstrated that the baseline characteristics of the retained children were not different from the entire population at baseline [**assuming that those dropped out were similar to those retained with regard to these characteristics**].” I have bracketed the concerning portion because in my view, that assumption renders the authors’ sensitivity analysis invalid.

What I am thinking of here is a sensitivity analysis in its simplest form as follows: those who were LTFU are one category and those who were never LTFU or missing are another. Then compare those 2 categories on things like LBW, malnutrition, preterm birth, infant sex, etc., What I see in the Supplementary table are descriptions of those who were retained at different time points, with no p values or comparisons. Just based on the numbers I see in the table, at 9 mo, the authors have 60/115 LBW kids (~52%) vs 515/792 normal birthweight kids (~65%). Which could very well be a statistically significant difference, but we don't know. The mean birthweight in your LBW children at 12 months was somewhat higher than in the LBW children at baseline. My point is that the children that were lost may be different in meaningful ways than the ones who were retained. My suggestion is to figure out whether there were differences, present p values in the supplement table to reassure the readers, and if there are differences, mention in the discussion whether you think they’d impact the results and how. It may be that you suspect that if everyone were there, you’d get even stronger associations.

Reply: We thank the reviewer for the very relevant comment and we agree with the concerns and we also believe that it is an added information that better helps interpret the findings. We have added some more details in the supplementary table like the percentages per line of those present at any specific time point compared to baseline. a statistical test was performed and the P values obtained have been added in the table. At 9 months, there are significantly more infants from the LBW group that did not attend that visit compared to the NBW group.

Minor comments and proofreading:

Line 35, don’t need comma after “both”

Reply: correction made, thanks.

Line 40 : included between 2009 and 2012 makes it sound like these are “n”s . Suggest you move the word “included” to after 2012.

Reply: the suggestion has been taken into account and the move made. the sentence now reads: “...conducted between 2009 and 2012, included infants…”

Line 48: It could be the same number of infants that were LBW, it’s just that you’ve lost more during follow up. What are the actual ns? Is it same kids?

Reply: Following the sensitive analysis, it appears that a significant number of LBW kids was missing at 9 months visit compared to the other group. The proportions remain similar at all the other time-points between LBW group and NBW group. This is now mentioned in the discussion.

Line 55: ratio, not ration.

Reply: typo was corrected

Body:

line 77 “”stunting is reported to be a strong marker of healthy growth”. It’s more like unhealthy growth, no?

Reply: that is true, typo was corrected

line 86” While stunting wasting and underweight…low birth weight…” maybe make that the first sentence of the LBW paragraph

Reply: Done

line 139: “like describe elsewhere”. This is not great grammar and not especially useful. I’d suggest you say something like “calculated using standard formula” and cite maybe DHS statistics or WHO web page that talks about how to calculate,

Reply: Sentence was reworded

line 139

Height-for-age Z-score (HAZ), weight-for-age Z-score (WAZ) and weight-for-height Z-score (WHZ) were used to define respectively stunting, underweight and wasting. HAZ <-2 standard deviations (SD), WAZ <-2 141SD and WHZ <-2 SD.

Suggest you reword to make complete sentences:

Ex: Height-for-age Z-score (HAZ, <-2 standard deviations below an international reference mean), weight-for-age Z-score (WAZ <-2 SD), and weight-for-height Z-score (WHZ <-2 SD) were used to define respectively stunting, underweight and wasting.

Reply: Sentence was reworded

144: data”were” . Data is a plural

Reply: word was corrected

Line 148: on comparing baseline characteristics of infants attending each visit to explore whether there was any alteration that could be imputed to the lost to follow-up.

Sentence is confusing. try instead “to explore whether any differences existed that could be associated with loss to follow up”

Reply: Sentence was reworded, thank you

Table 4: I’ve forgotten why Wasting is only assessed at month 1. I’d suggest the authors add a footnote to the table about that, because the authors obviously calculated prevalence of wasting at all times.

Reply: there were no preterm infants with wasting at 9 and 12 months, that ist he reason why point estimates are not shown at these time-points.

Reviewer #2: Most of my concerns have been addressed. However, it remains vague how the variables infant sex, maternal age and maternal literacy were selected for adjustments. What possible control variables did the authors considered at the beginning of the analysis? How did the end up in the three variables? This must be clearly described in the manuscript.

Reply: This is a good and relevant point raised by the reviewer. Actually, more variables were measured for the mothers during their pregnancies, these included parity, trimester of first antenatal clinic visit, BMI and MUAC and infectious status. These parameters have been controlled in the analyses of the determinants of LBW.

For the current analysis of the effect of LBW on growth, nutrition and mortality, the maternal variables age and literacy were considered to potentially affect the way mothers provide care to their infants and therefore any differences in these parameters could be confounding the potential effect of LBW only. The infant sex has been taken as a forced variable as sex and age are mostly considered as counfounding factors.

A clarifying sentence has been added in the methods section.

---

## [Editor Report · Decision Letter 2]

25 Jan 2021

Birth weight, growth, nutritional status and mortality of infants from Lambaréné and Fougamou in Gabon in their first year of life

PONE-D-20-19873R2

Dear Dr. Mombo-Ngoma,

We’re pleased to inform you that your manuscript has been judged scientifically suitable for publication and will be formally accepted for publication once it meets all outstanding technical requirements.

Kind regards,

Samson Gebremedhin, PhD

Academic Editor

PLOS ONE
---

## [Editor Report · Acceptance letter]

27 Jan 2021

PONE-D-20-19873R2 

Birth weight, growth, nutritional status and mortality of infants from Lambaréné and Fougamou in Gabon in their first year of life 

Dear Dr. Mombo-Ngoma:

I'm pleased to inform you that your manuscript has been deemed suitable for publication in PLOS ONE. Congratulations! Your manuscript is now with our production department. 

Kind regards, 

on behalf of

Dr. Samson Gebremedhin 

Academic Editor

PLOS ONE